# A Compensatory Response to the Problem of Evil: Revisited

**Michael Douglas Beaty**

Department of Philosophy, Baylor University, Waco, TX 76798, USA; michael_beaty@baylor.edu

**Abstract:** In this essay, I revisit the univocity thesis, Sterba's analogy between God and a leader of a politically liberal society, and, most fundamentally, whether the existence of horrendous evils is logically compatible with the existence of a good God. I concede that the typical appeals to free will and greater goods defenses to block the logical problem of evil are not sufficient because they do not adequately address the horrendous evils that are all too much a feature of human existence. While acknowledging that a compensatory response to the problem of evil is suggested by several important philosophers, I rely most centrally on the work of Marilyn McCord Adams. In so doing, I defend the thesis that the existence of a good God is logically compatible with the existence of horrendous evils, given God's capacity to absorb, defeat, or engulf it.

**Keywords:** compensatory response to the problem of evil; existential problem of evil; free will defense; greater goods defense; horrendous evils; playpen freedom; univocity thesis; Marilyn McCord Adams; James Sterba; Richard Swinburne

I am grateful that Professor Sterba has invited me[1] to continue our discussion about whether the existence of evil, in the amount and kinds of evil so prominently displayed in our world, establish the conclusion that the existence of a good God[2] is logically impossible. I am pleased to have the opportunity to "own up" to some mistakes, and clarify my argument, as well. Sterba agrees that Plantinga, and others, have shown that a good God is logically compatible with some moral evil. His contention is that the amount of evil in our world and its kinds, especially the horrendous evils that are so characteristic of our human history, are logically incompatible with a good God.[3]

## 1. Univocity Thesis and the Goodness of God

In my previous paper I claimed that both Sterba and I accepted the univocity thesis (at least with ascriptions of moral goodness to God) (Beaty 2021). He notes that he never employs this concept and emphatically rejects that he assumes it when speaking of God's moral goodness. In contrast, he says, "like the good Thomist I once was" we speak analogically when we make assertions about God and God's goodness. We do so "by analogy to features about ourselves and the rest of what is assumed for the sake of the argument to be God's creation."[4] I am unsure what he means, exactly, and how, given his arguments in his book, he is not committed to some form of univocity.

Clearly, discussions of the status of religious language in Theism is controversial, interesting, and, among some thinkers, such as Aquinas, complicated. For the purposes of this paper, I clarify my claims in the following way. Like Sterba, I accept that some of what we say about God presupposes some form of cognitivism in contrast to non-cognitivism with respect to the possible predication of moral properties. While it is true that theological and philosophical discussions of God's nature employ via *negativa* and analogies, we theists (at least some of us) sometimes attribute to God some moral properties that are best understood as being the same kind of moral property as possessed by human beings, even though God possesses them perfectly, rather than imperfectly. More importantly, I fail to see how one can get a variety of arguments from evil off the ground unless the moral concepts, and moral standards with which they are associated, are understood univocally when

applied both to human beings and to God. The various arguments from evil are supposed to demonstrate or make it probable that God does not exist, given the amount and kinds of evil in our world. For the argument to work, God must be assumed to possess properties which when taken together support the conclusion. Moral goodness is one such property (or set of properties). A natural reading of some of the Christian scriptures presumes God can be correctly described as possessing a bundle of moral properties which it is possible (and admirable) for human beings to possess, even if we human beings possess them, when we possess them at all, incompletely, imperfectly, or only partially.

The scriptures certainly speak of God as being compassionate, just, merciful and full of loving-kindness. Consider the following passage from Jeremiah 9:23–24:

> Thus says the Lord: "Let not the wise boast of his wisdom, and let not the mighty man boast in his might, let not the rich man boast in his riches; but let him who boasts, boast in this, that he understands and knows Me, that I am the Lord; who exercises loving-kindness, justice, and righteousness on earth; for I delight in these things, declares the Lord.

In Isaiah 58, God's prophet condemns the false worship of God and insists that the true worship of God includes

> . . . *loosening the bonds of injustice; . . . undoing the thongs of the yoke, . . . letting the oppressed go free, . . . sharing one's bread with the hungry, . . . bringing the homeless poor into your house;*

If I accept that God is speaking through the prophet Isaiah, asserting that God loathes injustice and loves us all with a steadfast love, then I accept that I have been provided grounds or reasons for claiming that God is just and loving and so on. In so doing, I am ascribing to God the moral virtues of being just and loving and compassionate and others besides. Are these ascriptions of moral goodness to God univocal or analogical? They appear to be ascriptions of moral goodness to God that are best understood as used univocally when compared to ascriptions of moral goodness to human beings. When the good kings and men and women of Israel loosen the bonds of injustice, release the oppressed to go free, share their bread with the hungry and welcome the homeless poor into palaces and homes, they are praised for their moral goodness. So, too, the God of Israel is morally good for God is on the side of the poor, oppressed and those treated unjustly. And God is at work via God's own initiatives to address the moral wrongs so characteristic of this world.

No doubt, if God exists as described or affirmed by traditional theists, then God possesses God's properties, to include moral properties, in ways that human beings do not (for example, necessarily versus contingently; perfectly rather than imperfectly). And Aquinas's rich and provocative analogical account of the theological attributes is motivated to capture the metaphysical differences, to be sure. But Aquinas' account of analogy, as I understand it, is not strong enough to support Sterba's argument. It appears to me that Sterba needs univocity for his argument from evil to have the force he takes it to have. If I am mistaken, I ask Sterba to make clear how his arguments that are meant to show the logical incompatibility of God and evil reflect an analogical use of attributions to God, rather than univocal ones.

Now there is a perfectly understandable way in which some of Sterba's ascriptions are analogical rather than univocal. In just politically liberal societies, like the United States of America or Great Britain, there are laws that prohibit unjust assaults. Additionally, there are agents whose professional job is to protect the freedom of its citizens, as far as possible, from various kinds of illicit uses of freedom. Just societies have executives whose responsibilities include the enforcement of laws that curtail the freedom of those who unjustly harm or kill their fellow citizens. God is similar to the chief executive of a politically liberal society, insists Sterba. Just as those who administer or govern good political states must restrict the freedoms of would-be aggressors to secure the more "significant freedoms" of other citizens (and their would-be victims), so a good God must restrict the freedom of human

agents. By reducing the illicit freedom of many of its citizens, God would be increasing the more important "significant freedom" of its citizens, argues Sterba.

Of course, here Sterba is employing an analogy when he compares God to the leader of a politically liberal society. Unlike anyone who serves as President of the United States, God is not elected to His office as Creator, Sustainer and Ruler of the Universe. If God exists, He occupies the office of Sovereign of the Universe, eternally, and not temporally. God did not begin to occupy this office after a democratically operationalized vote of the citizens of the universe. While both God and President Biden may be ascribed as sovereigns, their sovereign powers and responsibilities bear only analogies to one another.

It is correct, I think, to say that being a president of university is similar to being the president of the United States, though in a variety of ways, these two chief administrative roles are importantly different. Nonetheless, despite their significant differences one can draw analogies between the two administrative roles that may be illuminating and useful (or not). If we appeal to an analogy between God and President Biden's sovereign powers, how do we get the problem of evil off the ground? Typically, President Biden is not morally responsible for the morally impermissible actions of his democratic constituency. History rightly condemns many white governors of southern states for failing to do enough to promote and protect the freedoms of their African American citizens. Others are morally condemned for inciting their white constituencies to morally reprehensible actions and attitudes toward their African American fellow-citizens and neighbors. Is God morally at fault for the moral misdeeds of his constituency? For some of us, unlike Sterba, it is not obvious that God is at fault. And one part of our discontent is that Sterba's account rests on a flawed analogy.

Let us agree that if God exists as understood by traditional theists, then God and the President of the United States have at least some responsibilities that are analogous in nature. But when Sterba speaks of the shortcomings of either the free will or soul-making defenses (and the like) he, I claim, is not consistently appealing to analogy but assumes univocity. For example, consider the following moral principle identified by Sterba.

> Every moral agent has a reason not to interfere with the free actions of wrongdoers when permitting the slightly harmful consequences of those actions would lead to securing some significant moral good, in some cases, maybe just that of the freedom of the wrongdoers themselves, or to preventing some significant moral evil (Sterba 2019, p. 26).

He calls this the Principle of Noninterference or NI. By moral agent, clearly, he means someone whose actions can be correctly evaluated as either being morally required, morally permitted, morally forbidden, or morally indifferent. Moral evils are actions that are morally forbidden. These assumptions are necessary to make sense of NI. I claim that Sterba assumes univocity to prosecute his primary thesis:

> . . . the actual world we live in is such that there is much more that God could have done to promote significant freedom in it . . . Hence, the world we live in cannot be justified by the distribution and amount of significant freedom that is in it. There are too many ways that political states and human individuals could have increased the amount of significant freedom by restricting lesser freedoms of would-be wrong-doers. Likewise, there is much that God could have done to promote freedom by restricting freedom that simply has not been done (Sterba 2019, p. 29).

What the above paragraph suggests is something like the following argument: Argument A:

(1)　We rightly fault some political states and their human leaders because they failed to restrict in morally appropriate ways the freedoms of individuals who wronged their fellow citizens or fellow human beings.

(2)　If the theistic story is true, then God is like (the analogy) a human leader of a (liberal) political state.

(3)     If we rightly fault some (liberal) political state and their human leaders because they failed to restrict in morally appropriate ways the freedoms of individuals who wronged their fellow citizens or fellow human beings, then we ought to morally fault God for failing to promote (human) freedom sufficiently by more severely restricting the freedom of some human beings.

(4)     So, we ought to morally fault God for failing to promote human freedom sufficiently by more severely restricting the freedom of some human beings when they choose to act in morally evil or vicious ways.[5]

While argument A employs an analogy, its conclusion makes an assertion which does not employ analogical language. Its conclusion is best understood univocally, not analogically. To be sure, what it means for God to act or refrain from acting is a complicated metaphysical account. And, no doubt, it requires using some of our language in ways that stretch our ordinary meanings or uses of it. Nonetheless, it is unclear to me that Sterba is relying on analogy in formulating his logical argument(s) from evil (Thomistic account of analogy or otherwise) when he argues that we are in a position to see that affirming that the God of Theism exists is inconsistent with the existence of horrendous evil. Put more strongly, if Sterba is actually using language analogically, his argument fails to have the logical force it needs to establish the logically inconsistency of a good God and horrendous evil.

## 2. The Heart of the Matter and My General Strategy

The theme of Sterba's book is that he will use untapped resources from moral and political philosophy to show that the existence of evil of the kind and amount found in our world is logically incompatible with the existence of God. However, what is essential to his project is to successfully defend the following kind of argument.

Argument B:

(1)     If a good God exists, then there would not be the amount and kinds of evils (horrendous evils) that is characteristic our world.

(2)     Characteristic of our world is horrendous evils.

(3)     So, a good God does not exist.

Clearly the Argument (B) is formally valid. Is it sound? That is, are all the premises true? The critical premise is (1). Does the Theist have rationally credible reasons to reject premise (1)? I argue that theists do have credible reasons. First, a theist can appeal to the value of a world in which human beings are capable of courage or cowardice, just or unjust behavior, virtue or vice. Of course, to do so is to appeal to the familiar "greater-goods" defense. Second, the theist adds to the "greater-goods" defense an appeal to the "free will" defense. But these two, taken together are an incomplete defense of the claim that a good God is not logically incompatible with the existence of evil. More needs to be said, I contend. To address "what more needs to be said," I return to what I am calling a *compensatory response* to the problem of evil. The conjunction of these responses are logically possible. Taken together they constitute a satisfactory response to Sterba's attempted refutation of Theism, which was based on the existence of "horrendous evils." By "horrendous evils" I rely on Marilyn Adams who defines this category as follows: "evils the participation in (the doing of or the suffering of) which constitutes prima facie reason to doubt whether the participant's life could (given their inclusion in it) have positive meaning for him/her on the whole." (M. M. Adams 2006, p. 32).

Let me summarize what I take to constitute the essence of my response to Sterba, as outlined above. With respect to the "problem of evil":

(A)     We see the reasons that God allows some evils.

(B)     We recognize that there are some evils for which we do not see the reason for God to allow them, but we are unsurprised by our lack of cognitive access to all of God's justifying reasons for permitting such evils.

(C)     We see how all of them can be defeated.[6]

To explain and defend how horrendous evils can be defeated within the framework of traditional Christian Theism, I will appeal to several important themes found especially central to the work of Marilyn Adams.[7]

## 3. A Greater Goods and Free Will Defense: A Helpful Beginning

In Sterba's second chapter of his book, *Is a Good God Logically Possible?*, he argues that there is no successful free will defense (Sterba 2019, pp. 11–34; 209–21). His third chapter has as its goal to undermine a soul-making theodicy. It is important to remember that Sterba's goal is to show that the existence of a good God is logically incompatible with the existence of evil, which we typically identify as the logical problem of evil. It is important to remember that a theist does not need a successful theodicy to defeat the logical problem of evil. He or she need only show that the existence of a good God is logically compatible with the existence of evil, which must include the amount and kinds of evil we find in the actual world. In what follows, I adopt the strategy of Richard Swinburne, first by employing a version of the *Greater Goods Defense*, then, second, by appealing to the familiar free will defense.[8]

It is important to remember the differences between a theodicy and a defense. A theodicy aims to provide God's reasons for allowing evil, at all, and of the kinds and quantity of evils which are characteristic of our world. A defense merely claims that there are certain possible goods found in our world that provide God good reasons for permitting the evils characteristic of our world and of our human experience. An exponent of a successful defense need not claim that the goods identified which provide God reasons to permit the evils characteristic of our world are God's reasons for permitting the evils at issue. The theistic defender of God in the face of evil may be rightly skeptical about his or her grasp of God's actual reasons, yet, convinced that a good God, in fact, has good reasons.

Perhaps God wants to create some creatures that are able to freely worship God, or not, and to make other significant moral decisions about their own welfare, and the welfare of other creatures like themselves. On the one hand, this requires God to create a world which is predictable, rather than unpredictable—one that functions by stable regularities we may call natural laws. On the other, if God wants to encourage and promote and undergird stable good-making qualities which promote human flourishing (call these the moral virtues), and provide stable conditions for human beings to act from their moral duties, it is necessary that human beings have a range of morally significant actions.

Thus, it seems possible that a necessary condition of morally significant actions (those that are required, permitted, or forbidden; and of the moral virtues and vices) is libertarian freedom—a freedom to act not fully determined by antecedent causes. It also seems that it might be logically impossible for God to create human beings possessing libertarian freedom without the possibility of each human being acting badly, with respect to what is morally fitting (Swinburne 1998, "The of Moral Evil and Free Will", pp. 125–37; "Natural Evil and the Possibility of Knowledge", pp. 176–93). One way that libertarian freedom is made possible is that human beings experience desires for objects and activities that are morally blameworthy (Swinburne 1979, "Argument from Consciousness and Morality," pp. 152–79; "The Argument from Providence," pp. 180–99).

The previous paragraphs describe a world in which both natural evils and moral evils are possible.[9] Indeed, the possibility of natural evils like broken legs or arms and psychological harm, like the pain of a broken friendship, make possible certain kinds of moral goods such as prudence (practical wisdom), temperance, courage, friendship, justice, and temperance, as well as their correlative vices—imprudence, intemperance, cowardice, enmity, and injustice.[10]

Now the above is a part of a familiar strategy: the greater-goods defense. When we think of people we admire, among those we list are people who have exhibited great courage in the face of real and present dangers. Included on my long list are the following: Amelia Boyten,[11] Fannie Lou Hammer,[12] Martin Luther King, Jr.,[13] and John Kerry (Brinkley 2004). Boyten, Hammer, and King exhibited extraordinary courage during their opposition

and personal resistance to the practices of segregation in the South in the 1950s and 60s.[14] Former Senator John Kerry was an exceptional U.S. Navy Swift boat commander in the Mekong Delta during the Viet Nam War and, later, an antiwar activist. In both these roles and his role as US Senator, he exhibited significant levels of courage. The courage each individual exhibited was possible only because of physical, emotional, and social dangers, hence the possibility of both natural and moral evils.

First-order goods include pleasure, health, happiness, a virtuous life, friendship, personal and political freedom, and many other possibilities as well. Some first-order evils are pain, ill health, suffering, betrayal, death, slavery, and segregation. Second-order goods include sympathy, compassion, courage, perseverance, faith and a virtuous loyalty. Second-order evils include betrayal, cold-heartedness, cowardice, and disloyalty.

Most of us value our own individual freedom and we desire to live in social and political situations that encourage and promote civic life while protecting and preserving a wide range of individual freedom. At the same time a good civic life requires deterrents and penalties for those that inflict first order evils on us and our friends and fellow citizens. These deterrents or penalties will be second order goods.

### 4. The Alleged Refutation

In Chapter 2, "There is No Free-Will Defense," Sterba asks us to consider some familiar superheroes of our American culture such as Spider-Man/Peter Parker. Superheroes, like Spider-Man, are committed to limiting the freedom of would-be villains to protect other human beings from their vicious behavior. Sterba invites us to think about the case of Matthew Shepard, a gay man who lived in Wyoming. In a bar in Laramie, Wyoming, Shepard was befriended by two men who bought him drinks. Receiving an offer from them for a lift, Shepard was driven to a remote location, then they robbed, beat and tortured him. Hanging him on a barbwire fence, they left him to die. Discovered by a passing cyclist, Shepard later died in the hospital, never having regained consciousness.

Sterba notes that God could have intervened in this case in any number of ways and no one would have protested had God done so. In so doing, the freedom of Shepard's assailants would have been abridged, but the more important freedoms, such as freedom from being unjustly killed or unjustly injured and the like, would have been defended or secured. Indeed, had Spiderman intervened, we would be pleased and relieved. Of course, Spiderman is fictional. Yet, Sterba thinks our public affirmation of the role of superheroes is telling. We want someone with superpowers to intervene to save us or our neighbors when we are in danger. Sterba implies that if a good God exists, that good God would have intervened.[15] Since there was no such intervention, God does not exist, and the free will defense of God, given the existence of such shocking horrendous evils, fails. Let's put Sterba's argument more formally:

(1) If a good God exists, then a good God had a reason to interfere with the freedom of the two assailants of Matthew Shepard.
(2) If a good God had a reason to interfere with the freedom of the two assailants of Matthew Shepard and did not interfere, then a good God does not exist.
(3) So, a good God does not exist.

Clearly, this argument is not decisive. Suppose one accepts premise (1). Are you required to accept (2)? No, for God might have an overriding or mitigating reason for not interfering. If so, then (2) is false. I find the following account by Swinburne possible and compelling. It provides possible reasons that God did not intervene.

That human agents have libertarian freedom is a great good. Such freedom includes the capacity to do or to refrain from doing some act A. The capacity to engage in free action is a great good because it makes him or her " ... a source of the way things happen in the Universe." (Swinburne 1998, p. 84). Swinburne calls this way of acting or being in the world, a "responsibility for things." (Ibid., p. 101). Additionally, Swinburne insists on the value of "being of use," of helping, serving, sacrificing, that is, exercising our human freedom for good ends or purposes. (Ibid.)

In short, a good God has good reasons for creating a world with free creatures, such as human beings that have a responsibility for how the world is constituted, to some extent, both socially and physically. By so doing, God has created creatures that are capable of self-giving love, generosity, compassion, kindness; of courage, long-suffering, and humility; of fidelity in friendships, marriage, and so on. Clearly, a good God recognizes that free human beings may, on occasion, even frequently, act badly, morally speaking. We are capable of cowardice, vanity, cruelty, envy, lust, gluttony, infidelity, and a kind of anger that burns its objects and sometimes even its source(s).

Most of us, most of the time, regard human freedom as a great good and the ground of many other human goods, both moral and non-moral. When we Theists affirm the goodness of God, as Creator and Sustainer of the Universe, we affirm, among other goods, the good of human freedom. But, with this affirmation comes the recognition that we live in a world in which many human persons misuse their freedom to the great detriment of themselves, other human beings, other animals, and of other fundamentally valuable aspects of the natural world. Genuine human freedom, then, comes both the possibility of genuine moral goodness and a variety of moral evils.[16]

Suppose we generalize from the case of Matthew Shepard. On Sterba's view a good God would have intervened in Shepard's case and prevented the great evil of his horrific death. But there are many other similar humanly vicious acts. Is God to intervene in each and every case? We theists believe that God sometimes intervenes in our world to aid those in need. And often we pray for God's help, both in small and large matters, as the Biblical materials encourage and most Christian theologians endorse. Does this practice not undermine our belief that a good God exists? Not according to Swinburne, who says that

> "... in general, if God normally helps those who cannot help themselves when others do not help, others will not take the trouble to help the helpless next time, and they will be rational not to take that trouble. For they will know that a more powerful help is always available." (Swinburne 1979, pp. 210–11).

Yet, I concede (or confess) that the horrific evil visited on Matthew Shepard, despite its judicial resolution, may leave one theologically dissatisfied despite appeals to the good of free will and to the great good of soul-making. Is not the existence of a good God rightly called into question by the horrific suffering and the death of Matthew Shepard?

## 5. Sterba on the Soul-Making Theodicies and the Pauline Principle

Sterba contends the answer is definitively "Yes." He begins this chapter by reminding us that moral evils, especially horrific evils, cannot be justified by God's creating human beings with morally significant freedom (Sterba 2019, p. 49). If God is justified, it must be for other reasons. He begins his discussion of other reasons by considering soul-making theodicies. Does the opportunity for soul-making justify God's permitting the vast amount and kinds of evils our world contains? It does not, claims Sterba, because a large number of people fail to have an opportunity for significant soul-making in our world since their freedom is abridged by evildoers, a freedom evildoers should not have.

Sterba claims that there is an ethical principle that is in direct conflict with God's permitting evil. It is the Pauline Principle which claims that "one should never do evil that good may come of it." (Ibid., p. 49). Sterba concedes that there are exceptions to the principle which include (a) the harm caused is trivial or minor (I step on your toes in pushing you away from a speeding vehicle) or (b) when what I cause is easily reparable (while driving your car I swerve to avoid a pedestrian which results in an scratch on your left fender) or perhaps (c) it is the only way to prevent a much greater set of harmful consequences (one bombs a military target that is likely to cause significant civilian deaths but which will likely prompt surrender and much fewer military and civilian causalities) (Ibid., pp. 49–50).

Additionally, Sterba acknowledges another important objection. If God acted as Sterba insists he ought, then God provides human beings with what Christian theist Richard

Swinburne called "toy freedom" and atheist David Lewis called "playpen freedom".[17] Sterba comments that a playpen freedom would "greatly diminish our status as moral agents." (Sterba 2019, p. 52). Indeed, he says,

> Now no one doubts that there would be a problem if God always intervened to prevent evil. If that were to happen, then the freedom we would be left with would hardly be worthy of that name. Clearly, we must have freedom to do wrong if we are to develop through soul-making the virtue that would make ourselves less unworthy of a heavenly afterlife. But having the freedom needed for soul-making is not the same as having unlimited freedom ... Toy freedom or a playpen freedom is a problem only where freedom is constrained too much, not where it is appropriately constrained. But when are constraints on freedom too much and when appropriate? (Ibid., pp. 52–53).

So, Sterba concedes that it would be problematic if a good God intervened on each occasion of evil to prevent it. Human beings would have only a "toy" or "playpen" freedom. He appears to endorse the good of freedom as a necessary condition of soul-making as well. His objection to the appeal to soul-making as one part of a greater goods defense is this: to admit that human freedom is a necessary condition of soul-making is not the same as having unlimited freedom. To avoid toy freedom, we need not endorse unlimited freedom, so Sterba seems to conclude: We should expect God to intervene to prevent horrific evils, a qualitatively distinctive kind of evil, let's suppose.[18] Since horrific evils occur, God does not exist. But this follows only if it is reasonable to believe that a good God cannot have morally permissible reasons to permit horrific evils. I contend that Sterba has not demonstrated this to be the situation facing the theist. He needs to say more, but so do I.

### 6. The Just State, God and the Compensatory Response to Evil Introduced

Central to Sterba's argument that the existence of evil is logically incompatible with the existence of God is the ideal of what [the leaders of] a just and powerful politically liberal state do (or refrain from doing). In my previous paper, I objected by claiming that Christians do not, nor should they, regard the head of government of a politically liberal society as an adequate analogy for God's governance of the universe. Sterba's response:

> " ... here Beaty is not sufficiently taking into account the widespread use of analogy that compares God and Christ to an earthly king throughout the history of Christianity ... I just draw out the moral implications of this widespread use for the God of traditional Theism."

I agree that God and Jesus the Christ are often compared to earthly kings in the history of Christianity and in the biblical materials. When God is referred to as King of the Universe, God is being compared to an earthly King, not the democratically elected or appointed Chief of a political liberal state. Earthly Kings (or Queens) are not elected to their office by its citizenry. If the God of Theism exists, as the Supreme Being, God is neither elected nor appointed by the constituents of the universe. The implications of this disanalogy are important, and telling, in responding to Sterba's efforts to show that the existence of a good God is logically incompatible with the existence of evil. Unlike the Prime Minister of Canada or the President of the United States, God has all of eternity to reveal and display God's perfect moral goodness. This includes all of eternity to compensate those whose lives are vexed or shortened by horrific evils. I call this a Compensatory Response to the Problem of Evil.

Several important sources of this idea are available in the contemporary literature.[19] For example, in *Providence and the Problem Evil*, Swinburne says,

> ... God must choose to give each of us a life which is objectively in our best interest. ... it must remain the case that God must not cause harm to us which is uncompensated by benefit to us ... He must remain on balance a benefactor (Swinburne 1998, p. 232).

Eleonore Stump insists that "undeserved suffering which is uncompensated seems clearly unjust; but so does suffering compensated only by benefits to someone other than the sufferer." (Stump 1990, "Providence and the Problem of Evil," in (Flint 1990, p. 60)). And William Alston implies that God is committed to compensation for human beings when he asserts that "Any plan that God would implement will include provision for each of us having a life that is, on balance a good thing." (Alston 1996). But how is a good God to compensate the sufferer of horrendous evils? The short answer is that a good God must absorb those evils in such a way that the one who suffers is able to affirm his or her life, including the horrendous evils, as worth having. They do not "wish it away." (Adams 1990, p. 219).

Most impressively, this idea is developed and defended by Marilyn Adams in a variety of venues. In her "Horrendous Evils and The Goodness of God," she says,

> Where the *internal* coherence of Christianity is the issue, however, it is fair to appeal to its own store of valuables. For a Christian point of view, God is a being greater than which cannot be conceived, a good incommensurate with both created goods and temporal evils. Likewise, a good beatific, face-to-face intimacy with God is simply incommensurate with any merely non-transcendent goods or ills a person might experience. Thus, the good of a beatific face-to-face intimacy with God would *engulf* . . . even the horrendous evils humans experience in this present life . . . and overcome any prima-facie reasons the individual had to doubt whether his/her life would or could be worth living (Adams 1990, p. 218).

By "engulf", Adams means "a relation of organic unity between the negatively (positively) valued part and the whole, with the result that a significantly smaller negatively (positively) valued part can actually increase (decrease) the value of the whole of which it is a part." She contrasts "engulfing" horrific evils with "balancing off", a relation of value-parts to value-wholes. "Balancing off" is the kind of additive relation parts can bear to a whole within which one merely adds to a positively valued whole some negative part or vice versa (Adams 2006, pp. 45–46).

By engulfing the horrific evils individuals suffer, God ensures the one who suffers that his or her life is a great good to him or her on the whole. (Adams 1990, p. 218). According to Adams, this "engulfing" is possible

> . . . if we can offer a (logically possible) scenario in which God is good to each created person, by insuring each a life that is a great good to him/her on the whole, and by defeating his/her participation in horrors within the context, not merely of the world as a whole, but of that individual's life (Adams 1999, p. 55).

More will come on the logically possible scenario in the paragraph above in a moment. What is important at this juncture in the paper is to make clear the logic of compensation. In "Ignorance, Instrumentality, Compensation, and the Problem of Evil," Adams argues that while instrumental reasons have a place in moral practices and moral justifications, without non-instrumental reasons we run aground when we attempt to understand how a good God is compatible with horrendous evils. Clearly, we need some non-instrumental reasons that justify the claim that God is good to created persons. It is important to recognize that a good G for which some agent S allows E is often different from the good G* that compensates some agent T for the harm Y suffers because of E (Adams 2013, p. 17). Adams then insists that our biblical religion or faith proclaims that God is for us, not against us, and, thus, divine goodness will follow the logic of compensation. For those who have suffered horrific evils God will compensate him or her by guaranteeing that the life of each is a great good to him or her on the whole and in the end. This is possible only if all individual horrendous evils are defeated, overcome, absorbed in such a way that he or she would not wish away those evils experiences. Adams put it this way:

> Divine Love would permit horrors only if God could overcome them by integrating them into lives that are overwhelmingly good for the horror-participants themselves.

Adams maintains that engulfing is possible if the following propositions are logically possible: (1) God is the supreme, incommensurate Good; (2) God is Three Persons, but one God, thus instantiating a communion of persons; (3) God in Christ suffered horrendous evil through Christ's life, passion, and death; (4) We sufferers will be greeted with gratitude by the Triune God who suffered with us and for us; (5) None of those who are now enjoying the beatific union with God will retrospectively wish their sufferings away. I see no reason to think that the conjunction of these are metaphysically impossible.

In sum, according to Christian Theism, God suffered a horrendous death in the person of Christ. Christ's suffering and death opens a way to defeat the horrendous evils both Sterba and Adams have in mind. These horrendous evils that some human beings suffer in this life become a way of entering more fully into fellowship with God through Christ. If Christian Theism is true, the supreme good for human beings is fellowship with God, to include participation in the Triune fellowship. This communion with God is the highest good a human being can enjoy. Given this great good, even the horrendous evils some human beings suffer will be defeated by being engulfed by one's fellowship with God. In short, the objectively negative features of the horrendous evils some have suffered are defeated by being engulfed by the whole, which includes among its constituents, the great good of a deepened communion and fellowship with God and other human wayfarers.

## 7. Sterba's Rejoinder

Sterba's rejoinder to a compensatory response to the problem of evil includes a number of specific objections. Let me begin by quoting him.

> … that even given an eternal future it is not logically possible for God to adequately compensate for all the significant and especially evil consequences of moral action that God, if he exists, would have permitted in this world. Here, is why. First, God's restoring to exactly the way we were just before we were wronged by having the horrendous evil consequences inflicted on us in this life, which is the ideal of restorative justice, would never be better for us, given the lost time and opportunity the wrongdoing would entail, than God's preventing those consequences from being inflicted on us in the first place. Moreover, it may not even be possible for God to restore us to exactly the same way we were before we were wronged. Even God, it would seem, cannot erase the past. Second, any goods that are not logically connected to God's permission of horrendous evil consequences of wrongdoing would be goods that God could and should have provided without permitting especially horrendous evil consequences to be inflicted on us if he provided them to us at all. Third, for any goods that are logically connected to God's permission of horrendous evil consequences, the would-be beneficiaries of those goods would morally prefer that God had prevented the consequences rather than that the would-be beneficiaries be provided with those goods through God's permission of them.[20]

Sterba claims that it is not logically possible for God to adequately compensate the victims of the evils they have suffered. This is because:

(1) A never-sufferer is always better than a compensated sufferer due to the time or opportunities lost as a result of one's suffering.[21]
(2) It is not possible for God to fully compensate a sufferer because not even God can erase the past.
(3) So, God cannot compensate the sufferer fully.
(4) Any God-given "greater goods" not connected to any horrendous evils could and should be given without permitting horrendous evils.
(5) The would-be beneficiaries of these "greater goods" would morally prefer that God had prevented the consequences rather than that the would-be beneficiaries be provided with those goods through God's permission of them.

Let's begin with statement (1) which claims that "A never-sufferer is always better than a compensated sufferer due to the time or opportunities lost as a result of one's suffering."

First, Sterba's response presupposes that the conjunction of the greater-goods defense and the free-will defense, conjoined with a beatific union with God who is the Supreme Good for human beings, is a morally insufficient response to the logical problem of evil. Yet, this is an assertion, not an argument. How might one fill in the gaps to offer an argument that is consistent with Sterba's principle cited above? Let's distinguish between "ideal compensation" and "sufficient compensation."[22] With respect to "ideal compensation" let's stipulate that one receives exactly the goods lost for which one is to be compensated. In contrast, by "sufficient compensation" let's stipulate that one receives goods that sufficiently satisfy a person with respect the goods that one lost. While it is true that one who loses a limb in logging operation and receives a large financial settlement via his or his company's insurance has not received "ideal compensation", nonetheless, his compensation may be sufficient with respect to his present and future good(s). Ideal compensation is not a necessary condition for sufficient compensation.

In II Corinthians 11:23–33, we readers of this letter from the Apostle Paul learn of the many ways he has suffered, both mentally and physically, because of his being a faithful servant of Jesus the Christ. He was beaten, stoned, shipwrecked, imprisoned, and lashed, among other things. He went without food, sleep, and endured other physical and mental deprivations in order to preach what he believed to be God's good news. He recognizes that he is likely to die at the hands of the Roman or Jewish authorities. He seems to accept that he will be crucified, a horrific way to die while affirming that, in so doing, Paul will share in Christ's mode of death as a sign of his faith. Paul infers, thus, affirms that he will be compensated for his horrific sufferings and he is better for having suffered and been compensated. Clearly, Paul has in mind "sufficient compensation" and not "ideal compensation". In one New Testament passage he says,

> . . . we are afflicted in every way, but not crushed; perplexed, but not despairing; persecuted, but not forsaken; struck down, but not destroyed; always carrying about in the body the dying of Jesus, so that the life of Jesus also may be manifest in our mortal flesh. So, death works in us, but life in you . . . Therefore, we do not lose heart, though our outer man is decaying, yet our inner man is being renewed day by day. For the momentary, light affliction is producing for us an eternal weight of glory far beyond all comparison.[23]

While I have a hard time imagining myself desiring such a horrific fate, I don't think it impossible for someone to do so, and there seem to be many similar first person or third person testimonies of such similar confessions. The important point is that it is both logically possible and morally permissible for God to allow someone to suffer horrendous evils, given God's capacity to sufficiently compensate the sufferer, if God is a Supreme and Incommensurate Good.

Second, our grasp of them is incomplete, partial, and subject to error. Finally, we do not expect to know them, in any exhaustive way. In contrast, we Christian Theists insist that a good God can compensate those who suffer evils, horrendous or otherwise. In short, I presume that God has justifying reasons distinct from his capacity to compensate those who suffer horrendous evils, but I don't presume to be able to grasp those in any comprehensive and exhaustive fashion.[24]

Third, many philosophical and religious traditions affirm the value of suffering for the good. Clearly, Plato imagines that suffering for the good of the community and its civic life, as Socrates did, is not only noble, but the one who suffers in this way will receive more than adequate compensation. This compensation comes, either by virtue of its impact on one's own character or the way in which one's virtuous community honors the one who lived morally well. Perhaps, too, if there be life after death, in the afterlife one will receive compensation by being ushered into the presence of others that, too, have lived morally



noble lives or into the presence of those capable of compensating those who have suffered horrendous evils.

If it is morally fitting or commendatory for human beings to acknowledge and compensate their fellow citizens for such service, why is it morally better for God not to allow human beings to suffer at all rather than to compensate them for suffering when it is done to achieve morally noble ends? Doesn't this merely beg the question against traditional theistic understandings of the value of human free will, of soul-making practices, and of the possibility of union with God? In a variety of his letters, as suggested above, the Apostle Paul himself testifies to the fact that he has already been "compensated" for his service and fully expects a richer, deeper, and longer-lasting compensation. While these claims may be false, they are logically possible, unless on independent grounds Sterba knows that a good God—the incommensurately Supreme God—does not exist. But if he already knows that, he doesn't need a treatise whose thesis is that the existence of a good God is logically inconsistent with the existence of evil.

Consider statement (2): "It is not possible for God to fully compensate a sufferer because not even God can erase the past." First, if it is not possible that one who suffers be restored to exactly as he or she was before the person suffers, then God cannot be faulted for not doing so, just as God cannot be faulted for not making round-squares a constituent of our world. But perhaps this response is unfair to Sterba's point. Maybe the point is that because it is impossible to restore a person to exactly to the way he or she was prior to his or her experience of an horrific evil, neither the soul-making nor the greater-goods defense is an adequate justification of horrendous evils. But this claim just gives us an additional reason to think that these two defenses are not sufficient to adequately explain or justify horrendous evils. And, I agree. The difference between Sterba and me is this: Can God can sufficiently compensate the person for a real loss rather than provide an ideal compensation? My answer is "yes", which I defend below.

Second, and more importantly, let us concede that a person who suffers horrendous evils has lost much, with respect to finite and contingent goods, and these are real, substantial losses. Such real losses makes compensation morally important. Yet, since union with the Triune God is the greatest good, given the Christian story, despite God's not restoring the person to exactly where he or she was before he or she suffered, a union with God and other persons who share communion with God will more than compensates the person for their losses. Moreover, if the Christian story is true, the person so united to God and to other fellow believers will not want to be restored to "exactly as he or she was before suffering the horrific evils" because he or she will endorse their new state as better than the previous state.

At statement (3) Sterba insists that God cannot compensate those who suffer horrendous evils because God cannot restore "lost time and opportunity to the sufferer". If one places the emphasis on the quote, then it is another version of the "restored sufferer" is never as good as the "never sufferer". Another possible interpretation is to place the emphasis on a diminished moral capacity. If so, then that interpretation is unsatisfactory. Those who suffer are often at least as responsive, and often more responsive to the misfortunes of others, than those that have not suffered.[25] Indeed, some never-sufferers, do not suffer because they never take risks for the sake of a greater good. For example, they don't oppose racism or oppressive and evil regimes.

But, to return to the issue of compensation, if the Christian story is true, then God, the Supreme Good, has all of eternity to compensate the sufferer for having lost time and opportunity to secure certain goods constitutive or instrumentally valuable with respect to human flourishing. Thus, the "lost time and opportunity" objection is not an objection to God's capacity to provide sufficient compensation to those who suffer horrendous evils. When I forgo spending time with a family member during an anticipated vacation because it was necessary with respect to professional duties, it does not follow that I cannot make it up to that family member by some other activity or activities that satisfies, thus compensating him or her. In so doing, I have compensated them for the "lost time and

opportunity." If I can compensate for the loss of significant goods, surely, a good God can, but even more abundantly.

What about statement (4)? I confess that I don't grasp Sterba's objection. What are the "some goods" not connected to God's permission of horrendous evils? Among other things, Swinburne points out that God has a reason to create both an orderly world and a world which contains much beauty. Consider the state of Colorado. It is resplendent with snow-peaked mountains, some 14,000+ feet in height, arrayed in ponderosa pines and blue spruce forests, and drainage basins with beautiful rivers and creeks, which are bountifully housing a variety of trout species and other kinds of aquatic life. Every year there are numerous hiking, skiing or snowboarding accidents in the mountains of Colorado and rafting accidents on Colorado's numerous rivers or creeks. Every year these accidents result in some deaths or serious injuries. The goods connected with hiking these mountains in the summer and early fall, of fishing these rivers when the ice melt has ceased, and of skiing and snowboarding are many. In 2019, there were at least 109 deaths that occurred in outdoor recreation space in Colorado. Five years or so ago, I witnessed a death on the Arkansas River, near Buena Vista, Colorado, while fly-fishing with a friend. A raft with a family on board and its skilled rafter hit some rocks in a treacherous area and one of the family members was thrown overboard. The swift water and its undercurrent pulled him under, and his body was not recovered for quite some time. This is but one death, but for his family and friends, no doubt, his death was for them an horrendous evil. Surely, they have relived that incident often, wanting to rewind the script and have a different outcome. Is Sterba's point that a good God, if a good God existed, would intervene in every such incident so that no one would ever experience such an horrendous evil? Again, we get a "playpen" freedom, under this way of thinking, not a morally significant form of freedom. On the other hand, if a good God exists, a compensatory response is possible, though it requires, in part, a life after one's earthly existence. But, this is a part of the story Theists defend. Nothing Sterba has said shows this hope to be impossible. And what a grand hope it is.

Now consider statement (5). Perhaps there are two ways to understand (5). First, that would-be sufferers prefer not to have the good(s) that come about from such suffering to begin with. This implication is false, if the story about Thomas Broderick as told by Tom Brokaw is true in his *The Greatest Generation*.[26] Broderick is a part of a contingent of American and British paratroopers that parachute into Holland to take the Nijmegen bridge. Outnumbered by the Germans, on the fifth day, he got high in a foxhole and was shot, the bullet going clean through his temple. As I read the story, while it is true that Broderick, in one sense, would prefer that he not have been shot and blinded, in another, his life that follows his response to being blinded is so rich, he does not wish his blindness away. I see no reason to think his wish, if he in fact so wished, is psychologically impossible.

Second, perhaps Sterba is claiming that all human beings would prefer that God intervene in human affairs in such a way that no human being enjoy libertarian freedom so that horrendous evils are wholly eliminated. Why think this is true? Indeed, if the Christian story is true, the beatific vision enjoyed by those who suffer horrendous evils is different and richer than those who do not suffer horrendous evils. This is because they more fully enter into the "inner life of God" because God took on the horrors Himself. This is, in part, what Marilyn Adams means by her insistence of an organic relation between the horrors suffered and the compensating goods that engulf them. The person who "dies in the Lord" will be unified to God beatifically, whether or not they have suffered horrendous evils. Those who have suffered horrendous evils will enter more fully into the inner life of God because God took horrors into Himself, as the crucified and Risen Lord. Thus, oddly enough, I confess, the horrendous evils one suffered become a means by which one enjoys the greatest good more intensely.[27]

One can imagine that Saint Paul grasps these truths as he attempts to live faithfully in light of the life, death, and resurrection of Jesus, the Christ. In numerous passages in the letters of Paul, he celebrates the fact that he is suffering, as Jesus did, for the sake of God's

story of redemption for human beings. So, in a very important sense, Saint Paul did not prefer that God prevent the sufferings that he endured. He understood and accepted that his suffering is being overcome, defeated, and vanquished.

## 8. Compensation Revisited

In email correspondence with me, Sterba says,

"The real issue between us is the compensation issue. . . . The evil that God, if He exists, would have permitted is not necessary for any of us to have a decent life nor is it necessary for us to have the opportunity to be friends with God. We do not need the goods that are logically connected to God's permission of horrendous evils. An All-Good, All-Powerful God, if He exists, would have prevented this evil whose goods we do not need. The idea that God can adequately make up in the afterlife for permitting horrendous evils, the would-be beneficiaries of which would morally prefer that he had prevented—does not make moral sense. It is like saying that Dickens's Scrooge is perfectly good because he changed his ways near the end of his life. What we have in Scrooge is a character, who, even while he does now good things at the end of his life, would have wished he had not done what he had done earlier in his life. The God of traditional theism cannot be like that."[28]

The following points are made in the text of his email quoted above:

(1) The evils that God permits are not necessary for any of us (significantly free human beings) to have a decent life.
(2) Nor are they necessary for us to have an opportunity to be friends with God.
(3) We do not need the goods that are logically connected with God's permissions of horrendous evils.
(4) If a good God exists, then a good God would have prevented the evils for goods we do not need.
(5) That God can actually compensate those who have suffered horrendous evils does not make "moral sense". It is like saying God is perfectly good because he changed his ways near the end of his life while admitting that God wished God had not done what God did at some earlier point in earthly time.

While endorsing the value of human freedom, the traditional theist agrees that the moral evils God permits are not necessary for human beings to have a decent life. However, these evils are the result of human beings misusing their freedom. The natural evils that God permits are a consequence of our finite and vulnerable natures and the world which God has given us to inhabit. It is a good world, we confess, but much of its natural evils are a function of its being one in which our freedom matters since it is possible for us to harm ourselves, others, and the world in which we live. The alternative posed by Sterba is, I suggest, once again, a "play world."

Our friendships with one another matter in a variety of ways, and in varying degrees. Whether we are speaking of the kind of friendship that Aristotle rightly calls, "another himself" (Aristotle 1985, 1166a, 30–33) or those to whom we are amiable and kind, but much less intimately related, or those with whom we are friends, in so far as share a common community or working environment, it is possible to either treat that relationship too lightly, or to treat it callously, or to betray it altogether. Sterba has given no reasons to think that being friends with God requires an entirely different set of attitudes and modes of relating. In the Gospel stories, Judas' betrayal of Jesus is surprising and vicious precisely because Judas had exercised his freedom to join Jesus' closest circle of friends, which included a kind of intimacy and trust for which betrayal is a great evil. The Jewish and Christian scriptures emphasize that the goods of friendship with God, Jesus the Christ, and his genuine friends, are costly. In Sterba's world, there are no such costs. Given the absence of these costs, then, moral praise, moral admiration, and moral revulsion are not possible.

Sterba insists that we do not need the goods that are logically connected with God's permissions of horrendous evils. What are those goods, the goods connected with horrendous evils? As far as I can see they are, first, the ordinary human goods of physical and emotional well-being, grounded in food, clothing, shelter, the kind of fair and consistent application of law and order that preserves and promotes them. Sufficient conditions for them include a just and stable social order that promotes respect for fellow citizens and members of other social and political communities. And, secondly, that human beings have a libertarian freedom by virtue of which each individual becomes virtuous or vicious by freely aligning himself or herself with good rather than evil. Important commitments of politically liberal societies are a wide range of political freedom grounded in the presumption of individuals having libertarian free wills. When Sterba insists that we do not need the goods that are logically connected with God's permissions of evils he is insisting that if there were a God, we would not be significantly free, just so that the evils that vex our world would be absent. I reject his claim while I know that some theists do indeed hold that what happens in the world is the only thing that could have happened because God has decreed this be so in every detail. If human beings are significantly free and God does not intervene at every moment for the sake of the morally better, then moral evil will occur, given that there are some, indeed, a great many, vicious moral agents. And at times we cry out, "Enough, O God. Make it go away, altogether, and by Your actions, O Lord, alone!"

The uncompromisingly honest, keen-sighted theist is no naïve, besotted optimist. He or she is moved by the suffering of the world, its rampant evils and injustices as, indeed, Sterba is. It is precisely at this point, the issue of compensation for those who have suffered unjustly is morally relevant. But of the conception of God who compensates those who have suffered horrendous evils Sterba asserts: "It" [God's compensating victims of horrendous evils] makes no moral sense." This is because

A.   It is like saying Scrooge is perfectly good because he changed his ways near the end of his life while

B.   Admitting that God wished God had not done what God did at some earlier point in earthly time.[29]

However, I don't see the view that God's goodness includes compensating those who have suffered as subject to either of these possible shortcomings, thus, being morally, or otherwise, less perfect. I think there are a number of reasons to reject Sterba's assertion, many of which I have already identified and defended. But here is one more effort to that end.

One ordinary understanding of compensation is the activity of providing someone some significant good for the loss of some other significant good. Often we associate compensation with being provided money for a work-related physical or psychological injury. But the consultant that reviews the practices of an academic department, a family-owned business, or a public hospital and receives payment for his or her consultation is being provided some significant good for their work and the time it takes to do the work well. A significant good need not be money, however. When in a pinch, a valued colleague does more than is required for the good of the department, and in doing so, spends less time on his or her own projects or in promoting or maintaining the good of his or her family, it is appropriate to recognize and to honor his or her sacrifices. In doing so, some compensation is provided the one making the sacrifice. Given this sort of example, one can imagine the compensation taking a wide variety of forms. One might receive a monetary bonus from one's employers as a means of compensating the valued employee. Or he or she might receive public recognition in a forum and in a manner both the employee and his or her colleagues will value. It is a fundamental conviction of the Theistic religious traditions that God desires fellowship with human beings and that human beings are created for this as a constituent of their actual flourishing. Sterba has given no reason to think that it is logically self-contradictory to suppose that a good God can and will compensate the victims of horrendous evils, superabundantly, by fellowship with God. So, in contrast to Sterba, I assert that the claim that God compensates human beings for the evils, horrific

or otherwise, makes sense. It is morally and conceptually intelligible. And union with God will superabundantly compensate human beings for any and all horrendous evils they suffer.

## 9. Conclusions

I claim that Sterba has not shown that the existence of a good God is logically inconsistent with the existence of horrendous evil. I hope I have succeeded in this intellectual task. Even if I have, I know full well that I have not solved the problem of evil as a lived experience. There is more to the problem of evil than the puzzle it provides our intellects. For those who have close friends or important family members who have died, tragically or unexpectedly, the death of one of them often causes an emotional or existential crisis that I refer to in an unpublished essay, the "existential problem of evil". Douglas Gresham, C. S. Lewis' stepson, says of Lewis' written response about the death of Graham's mother and Lewis' wife's (Joy Davidman), that the book was "one man's studied attempts to come to grips with and in the end defeat the emotional paralysis of the most devastating grief of his life." (Doug Gresham 2001, "Introduction" in C. S. Lewis, p. XXI). Two other books of the same genre are Nicholas Wolterstorff's *Lament for A Son* (Wolterstorff 1987). and Paul Wisely's *Keeping Up The Heart: A Father's Lament for His Daughter* (Paul W. Nisly 1992). These books are insightful but painful reflections on death and grief, written by grieving Christian wayfarers who did not expect their sorrow to go quickly away, either into that dark night or that bright noontide day. Yet, each author, like C. S. Lewis, affirmed that their faith in a good God as something, ultimately, both intellectually and morally fitting. I join them in that affirmation.

**Funding:** This research received no external funding.

**Conflicts of Interest:** The author declares no conflict of interest.

## Notes

1  For the initial essay, see (Beaty 2021). In addition to Dr. Sterba's invitation to continue our conversation, I am grateful for the constructive comments and helpful suggestions on this paper provided by Baylor graduate students, Mr. Nick Hadsell and Ms. Kelsey Maglio, and Dr. Todd Buras (chair), Department of Philosophy, Baylor University.

2  By 'good God' Sterba means the traditional Theistic affirmation of God as a being that is maximally perfect in knowledge, power, and moral goodness.

3  By good God, Sterba means the omni-God of traditional Theistic affirmations and recent discussions in analytic philosophy of religion. Richard Swinburne's definition of the Theistic God is: a person without a body, who is eternal, perfectly free, omnipotent, omnisicient, perfectly good, and the creator of all things. See (Swinburne 1979, p. 8).

4  Sterba, his response to Beaty in *Religions* (Sterba 2021).

5  The argument posed here is not articulated, as such, by Sterba. It is consistent with his main theses and captures the force of his objections, I contend.

6  In a recent discussion with me about the problem of evil, this succinct summary was articulated by Dr Todd Buras, chair of the department of philosophy, Baylor University.

7  See especially (Adams 2013, pp. 7–26; Additionally, Adams 1990, pp. 209–21; Adams 1999, pp. 155–80; Adams 2006, pp. 53–79).

8  While many theists have provided responses to atheistic arguments from evil based on an appeal to soul-making and free will, I am primarily relying on the work of Richard Swinburne.

9  Clearly, the material in these paragraphs is not original with me. (Swinburne 1979, pp. 180–99) and (Swinburne 1998, pp. 125–219).

10  As an example of goods and evils I point to the four cardinal virtues and their correlative vices. But there are many other virtues and vices made possible by libertarian freedom.

11  https://en.wikipedia.org/wiki/Amelia_Boynton_Robinson (accessed on 8 August 2022).

12  https://www.womenshistory.org/education-resources/biographies/fannie-lou-hamer (accessed on 8 August 2022).

13  https://en.wikipedia.org/wiki/Martin_Luther_King_Jr (accessed on 8 August 2022).

14  Born in 1950 and raised in Benton, Arkansas, I vividly remember the Little Rock Central High School "crisis" generated by the "Little Rock Nine." See https://www.history.com/topics/black-history/central-high-school-integration (accessed on 8 August 2022).

15    Clearly, Sterba thinks of God as analogous to a superhero. But this assumption is inconsistent with a correct theological understanding of the Christian (Jewish and Muslim) God. In short, this assumption is a bad analogy.

16    No doubt, like other readers of this essay, I lament the Russian invasion of Ukraine and the loss of life and property this war includes.

17    (Sterba 2019, p. 52). On this topic, also see (Swinburne 1998, pp. 242–43). Additionally, (Swinburne 1979, p. 219).

18    Let us accept that "horrific evils" is primarily about qualitative distinctive evils which are also displayed in quantativeily disturbing amounts.

19    Among those are a blog by my colleague, Dr. Alex Pruss. See (Beaty 2021).

20    Sterba, "Response to Beaty" in *Religions* (Sterba 2021).

21    This is a more succinct and pithier restatement of one of my initial premises. I am grateful to one of my reviewers for this suggested revision.

22    I am grateful to an anonymous reviewer for prodding me to make this distinction.

23    2 Corinthians 4: 8–12; 16–17. *New American Standard Bible: Inductive Study Bible*.

24    My thanks to Dr. Todd Buras—colleague, friend, and chair of Department of Philosophy, Baylor University—for helping me see this point more decisively.

25    See the story of Thomas Broderick, shot in the head and, thus, blinded in World War II, told by Tom Brokaw in his *The Greatest Generation*. After spending some time angry about his fate, Thomas got on with his life, both marrying, having seven children, and establishing a successful insurance business. Broderick and some of his friends established a Blinded Veterans Association so he could share the lessons of his new life with other veterans struggling with blindness. A Catholic Christian, once he got over his initial anger, he set out to be the best husband, father, businessman, and citizen he could be . . . " (Brokaw 1998, p. 24)

26    See the previous note.

27    A recent conversation with Dr. Todd Buras – colleague, friend, and chair of Department of Philosophy, Baylor University—who helped me grasp this possibility point more clearly via several discussions of Marilyn Adams' contributions on these topics.

28    Sterba, email to me on 17 May 2021.

29    This was in an email correspondence with me.

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
