# Peer review of "A Compensatory Response to the Problem of Evil: Revisited"

_religions, doi:10.3390/rel14010035_

Round 1

Reviewer 1 Report

The paper deals with a classical question of theism. In doing this, its originality and innovation are poor. On the other hand, the argument is clearly presented and well-shaped. In the conclusions, the author frankly admits the limits of a pure theoretical reflection when existential grief is involved.

It remains unclear when and where prof. Sterba has invited the author to discuss the issue. I suggest to clarify this point, or simply to delete the first lines of the paper.

Reviewer 2 Report

Be careful in future submissions to properly prepare for blind review. There are clear evidences of authorship contained within the piece.

The essay is longer than it needs to be. Some things can and should be consolidated. Other things should be taken out altogether (e.g., the excursus on "the univocity thesis"). The central issue boils down to the compensation defense and the extent of its relevance. This is where the author's defense is too weak. For, compensation ought not to be the only consideration in the world God would create. Perhaps one suffers so that *others* benefit or that other necessary events obtain by which even to have a human race. And the idea that no evil should obtain no matter what the benefit of it is and which sufferers benefit has no independent support. But Sterba is not called out for this. Instead, the author plays too much defense here by citing Scripture and Marilyn McCord Adams about how such goods brought about through such evils are superlative. That's an okay point to make, but it must be made in conjunction with (i) that the evils that get one there are necessary, and (ii) it's such an incommensurable good that others should be able to acquire it. If horrendous evils are not permitted, not only would there be such superlative goods, but there might not even be a decent life for anyone. As John Hick says, while a world without such evils might be suited for a hedonistic paradise, as a world involving growth, maturity, knowledge, and ethics, it would be the worst possible world.

In terms of prose, spelling, and grammar, there are far too many issues to exhaustively point out all of them without spending a great deal of time. I've tried to highlight about 80%. The paper would benefit from a serious second proofread and needs to be, overall, structured better.

I don't think I can recommend this for publication. I have done my best to help the author in offering line-by-line critiques and suggestions (both content and mechanics), but at the end of the day it is at best only an inchoate work that hasn't quite landed. As such, it might augur well as a reply in a different venue.

Reviewer 3 Report

I think the introductory paragraph can be approved - 

A few minor corrections are needed:

Line 18 -  “that Professor Sterba has invited me” endnote doesn’t match

Line 27 -  In my previous paper I claimed”   -include reference to paper

Line 122 – “others are morally”   - capitalize O in others

Line 174 – “relying on analogy in formulated his logical” – should be "formulating"

Line 251 – “in fact, as good reasons.”   "As" should be "has"

Line 370 – ” He begins this chapter”  reference to work should be included

Line 574 – “that he will crucified” insert “be”

Line 778 – “ordinary language understanding of compensation”  delete "language"?

Round 2

Reviewer 2 Report

The restructuring and reformulating of key points in the paper have made this a force to be reckoned with. There are a couple of inset comments involving a revisit of some of the concerns to help further augment the paper (see attached). It's not necessary (of course) that any or all of them be adjusted, but they do attend certain unnecessary vulnerabilities that remain in the paper. And some residual minor grammatical/spelling and expressions are noted as well for consideration.

Author Response

First, the reviewer faults me, correctly, for accusing Sterba of question begging when he claims the never-sufferer is always better than the compensated sufferer.  He claims the latter because of "lost time and/or opportunities".  To address Sterba's real point, I make a distinction between "ideal compensation" and "sufficient compensation" and concede God's response to horrendous evils is in theory a sufficient compensation, even if not clearly ideal, given that lost opportunities and lost time cannot be fully recovered.

Secondly, the reviewer points out, correctly, that in another portion of the paper I initially respond to Sterba's claim that "it is not possible for God to fully compensate a sufferer because not even God can erase the past" by comparing this to not being able to make a round-square, so God cannot be faulted.  A more fair reading of Sterba's point is to treat his criticism as a moral reductio and respond to that criticism.  I leave in "God cannot be faulted for not making round squares" but more fully develop my response to Sterba around the distinction between "ideal compensation" and "sufficient compensation".  In short, God cannot be morally faulted for arranging the world in such a way that ideal compensation is impossible.  Morally speaking, "sufficient compensation" suffices.  
